# Complementation of an Eisosomal Yeast *pil*1 Mutant and Characteristics of Eisosomal Distribution in Hyphae of *Neurospora crassa* Germinating from Two Different Spore Types

**DOI:** 10.3390/jof9020147

**Published:** 2023-01-22

**Authors:** Krisztina Kollath-Leiß, Qin Yang, Hannes Winter, Frank Kempken

**Affiliations:** Botanical Genetics and Molecular Biology, Botanical Institute, Christian-Albrechts-Universität, 24118 Kiel, Germany

**Keywords:** eisosome, LSP-1, PIL1, *Neurospora crassa*, CLSM

## Abstract

Eisosomes are plasma-membrane-associated protein complexes of fungi and algae involved in various cellular processes. The eisosome composition of the budding yeast is well described, but there is a limited number of studies only about eisosomes in filamentous fungi. In our study, we examined the *Neurospora crassa* LSP-1 protein (NcLSP1). By complementing a *Saccharomyces cerevisiae* Δ*pil*1 mutant strain with *nclsp*1, we show the functional homology of the NcLSP1 to yeast PIL1 rather than to yeast LSP1 and hereby confirm that the NcLSP1 is an eisosomal core protein and suitable eisosomal marker. The subsequent cloning and expression of the *nclsp1*::*trfp* reporter gene construct in *N. crassa* allowed for a systematical investigation of the characteristics of eisosome formation and distribution in different developmental stages. In *N. crassa*, the hyphae germinating from sexual and asexual spores are morphologically identical and have been historically recognized as the same type of cells. Here, we demonstrate the structural differences on the cellular level between the hyphae germinating from sexual and asexual spores.

## 1. Introduction

The plasma membrane of fungal cells is of lateral heterogeneity [1] and allows for the spatial segregation of different domains with various functions [2]. Among them, MCCs (membrane compartments occupied by CAN1) are nanoscale, furrow-like invaginations at the plasma membrane, associated with spatially stable protein complexes called eisosomes [3]. Eisosomes were first proposed and described in detail in the yeast *Saccharomyces cerevisiae* [4,5], followed by studies on *Aspergillus nidulans* [6,7], *Candida albicans* [8,9], and *Beauveria bassiana* [10].

In *S. cerevisiae*, MCC domains are defined as 300 nm long and 50 nm deep furrow-like invaginations of the plasma membrane [3,11,12,13]. Two integral membrane proteins, SUR7 and NCE102, build the core of the domain [12,13,14]. On the cytoplasmic surface of the invaginations, protein complexes called eisosomes are found [15]. Eisosomes consist of several facultative and obligate protein components. The most prominent eisosomal core components in yeast are PIL1 and LSP1 [15]. The two BAR-domain-containing proteins with an amino acid sequence similarity of 74% self-assemble to heterodimers, which are responsible for the membrane bending at the MCC site [15,16,17,18,19]. The phosphorylation of PIL1 and LSP1 by the PKH1/2 kinases is crucial for correct eisosome formation [20,21]. Besides its eisosomal localization, *S. cerevisiae* PIL1 also accumulates in a cytoplasmic pool [18]. A large coiled-coil peripheral membrane protein without recognizable functional domains, SEG1 (stability of eisosomes guaranteed), is required for the efficient incorporation of PIL1 into eisosomes [4,22,23]. *Δpil1* mutants show the clustering of MCC furrows with less but larger eisosomal complexes [5].

The functional properties of eisosomes are not yet well understood [13]. Eisosomes play a role in a variety of cellular processes, such as plasma membrane organization, cell wall synthesis and morphogenesis, sphingolipid homeostasis [11,24], and stress response [11].

In filamentous fungi, eisosome composition seems to be similar but not identical to its *S. cerevisiae* counterpart. In *Aspergillus nidulans* and *Aspergillus fumigatus*, the two proteins PILA and PILB exist following a gene duplication event. Both are homologs to the *S. cerevisiae* eisosomal proteins PIL1/LSP1 [11,12], with PILA being functionally more convergent to PIL1 and showing a much higher similarity [6]. There are also differences in eisosome formation in different developmental stages between yeasts and filamentous fungi. In the filamentous ascomycete *A. nidulans*, PILA is localized in punctate structures at the plasma membrane in all developmental stages, and PILB and SURG (homologs to the *S. cerevisiae* SUR7) colocalize with PILA to form complete eisosomes exclusively in asco- and macroconidiospores [6,7]. Eisosome formation during asexual development has been systematically described in *Ashbya gossypii* [23]. Eisosomes are continuously present in this yeast-like filamentous fungus, with higher density in early germination stages compared with mature hyphae. New eisosomes are formed at the first 30 µm of the hyphal tip, excluding exocytosis sites, with a rate of 1.6 (+/−0.5) eisosomes per minute. In *Neurospora crassa*, the LSP1 (NcLSP1) protein represents a homolog to *A. nidulans* PILA. Hence, it putatively is functionally more similar to *S. cerevisiae* PIL1 than LSP1. The NcLSP1 is continuously localized to eisosomal patches at the plasma membrane [13,25]. Investigations in *N. crassa* show that eisosome formation is influenced by the cytoskeleton [26].

In our study, the functional convergence of the *N. crassa* LSP-1 to *S. cerevisiae* PIL1 protein was investigated through heterologous complementation to confirm the NcLSP1 as a suitable eisosomal marker protein. In subsequent experiments, the NcLSP1 was fused to a fluorescent protein to investigate the characteristics of eisosome formation and localization in different developmental stages and the hyphal types of *N. crassa*.

## 2. Materials and Methods

### 2.1. Strains

*N. crassa* strains used in our study are listed in Appendix A. All expression-vector-carrying fusion constructs were transformed into the histidine auxotrophic strains FGSC #6103 (his-3 (Y234M723) mat A) and FGSC #9716 (his-3 (Y234M723) mat A) provided by the Fungal Genetics Stock Center (FGSC; Kansas City, MO, USA).

The *S. cerevisiae* strain (*ade2-1*, *can1-100*, *his3-11*,*15 leu2-3*,*112 trp1-1*, *ura3-1*, *LSP1-GFP:HIS*, and *pil1Δ:Kan*) used for the yeast complementation experiment was kindly provided by the Farese and Walther Lab [5]. *S. cerevisiae* Y187 (*MATα*, *ura3-52*, *his3-200*, *ade2-101*, *trp1-901*, *leu2-3*, *112*, *gal4Δ*, *met–*, *gal80Δ*, and *URA3::GAL1*UAS *-GAL1*TATA*-lacZ*) (Clontech, Mountain View, CA, USA) was used as a non-fluorescent control.

The bacterial strain *E. coli* XL1-Blue (*recA1*, *endA1*, *gyrA96*, *thi-1*, *hsdR17*, *supE44*, *relA1*, and *lac F’proAB lacIqZΔM15 Tn10 (Tetr)*) (200249, Stratagene) was used for the propagation of vector constructs.

### 2.2. Media and Growth Conditions

Vogel’s minimal medium with 2% sucrose (VMM + S) was used for the cultivation of *N. crassa*, and Vogel’s minimal medium with 1% sorbose, 0.05% glucose, and 0.05% fructose (VMM + SGF) was used for single colony selection on the plates. *N. crassa* microconidia production was induced on a synthetic cross medium (SC) supplemented with 0.06% iodoacetate [27,28]. Westergaard’s medium was used for the crossing experiments [28]. The media prepared for the *his*-3 *N. crassa* strains FGSC #6103 and #9716 were supplemented with 0.02% histidine (VMM + S + His). Fungi were cultivated in a climate chamber at 25 °C under long-day conditions. For germination analyses, asexual and sexual fungal spores were cultivated at 30 °C for 5 h.

The *S. cerevisiae* strains were grown in a YPD medium (2% peptone, 1% yeast extract, and 2% glucose) at 30 °C and 200 rpm. Specific supplementations are indicated in the corresponding sections.

The *E. coli* strains were cultivated in a Luria Broth culture medium supplemented with the required antibiotics at 37 °C and 250 rpm.

### 2.3. DNA and RNA Isolation

For DNA isolation, the *N. crassa* strains were cultivated at 25 °C for 3–4 days, and the mycelium was ground under liquid nitrogen and transferred into lysis buffer (10 mM Tris-HCl, 1 mM EDTA, 100 mM NaCl, 2% SDS, and pH 8.0). DNA extraction was performed with phenol. The aqueous phase was then incubated with 100 µg RNase A, followed by additional phenol extraction and ethanol precipitation steps. Bacterial plasmid DNA was isolated using a NucleoSpin Plasmid Easypure Kit, according to the manufacturer’s recommendations (Macherey-Nagel, Düren, Germany). Plasmid isolation from yeast was carried out according to standard procedures [29].

RNA isolation from 3–4-day old mycelium was performed with peqGold TriFAST (Paqlab, Erlangen, Germany), according to the manufacturer’s instructions, with a subsequent DNase treatment step using DNaseA according to the standard protocol.

### 2.4. PCR and RT-PCR Amplifications

cDNA synthesis was carried out with a one-step RT-PCR Kit (Qiagen), according to the standard protocol.

PCR was performed as described previously [30]. MolTaq (Molzym), Q5 High-Fidelity (NEB), and *Pwo* DNA polymerases (PeqLab) were used for DNA amplification. The primers used in our study were synthesized by Eurofins MWG Operon (Ebersberg, Germany) and are listed in Appendix A.

Agarose gel electrophoresis was performed as described previously [31]. DNA markers from MBBM (Bielefeld, Germany) and NEB (Ipswich, MA, USA) were used to determine DNA fragment size.

### 2.5. Plasmid Construction

All the vectors used in this study are presented in Appendix A. The construction of the plasmid pJQ771 (used for creating the strains NcT462 and NcT475) carrying the *N. crassa lsp1::rfp* reporter gene construct under the control of the strong *ccg1* promoter has been previously described [25]. The plasmid pQY868 (used for creating the strain NcT507 and NcT508) also carries the *N. crassa lsp1*:*:rfp* reporter gene construct but under the control of the *N. crassa lsp1* native promoter (amplified with the primers QY2492 and QY2493).

For the yeast complementation experiments, the plasmids pHM893 and pKK904 were created. *N. crassa lsp1* cDNA was amplified by the primers QY3213 and KK2348, and *S. cerevisiae* native *pil1* was generated using the primers KK3488 and KK3489. Both PCR amplicons contain the BamHI and BglII recognition sites. The *trfp* ORF [32] was isolated from the vector pJQ771 by KK2350 and KK2349, containing the recognition sites BglII and SwaI, respectively. Blunt-end PCR products were subcloned using a CloneJet PCR cloning Kit (Thermo Fisher Scientific, Waltham, MA, USA) according to the manufacturer’s recommendation. The yeast expression vector pYES2/NT A (Invitrogen, Carlsbad, CA, USA) and the subcloned fragments were digested by the corresponding restriction endonucleases from NEB (Ipswich, MA, USA), according to the standard protocol. Gel elution was performed with NucleoSpin Gel and a PCR Clean-Up Kit (Macherey-Nagel), according to the manufacturer’s recommendations. Ligation was performed with T4 DNA-ligase by NEB (Ipswich, MA, USA) according to the standard protocol.

All final vectors were verified by sequencing. All the oligonucleotides used in this study were synthesized by Eurofins Genomics (Ebersberg, Germany) and are listed in Appendix A.

### 2.6. Transformation and Transformant Selection

Previously described methods were used for *E. coli* [31] and *N. crassa* [33] cloning and transformation via electroporation. The transformation of yeast was performed via electroporation, according to the methods formerly published [25]. Putative transformants were selected on a 2% glucose-containing solid minimal medium without uracil (SC/-Ura) at 30 °C. The expression of the desired protein was induced in a 2% galactose-containing fluid minimal medium at 30 °C and 180 rpm for 48 h. The expression was monitored via epifluorescence microscopy. All the cloning and transformation experiments were conducted in accordance with the requirements of the German gene technology law (GenTG).

### 2.7. Immunolocalization

The verification of the expression of the full-length LSP1::tRFP protein in *N. crassa* was carried out using an immunodetection assay (results Appendix A). The total proteins of *N. crassa* strains NcT507 and FGSC #6301 were isolated from germinating conidiospores. The liquid minimal medium was inoculated with conidia and incubated for 5 h at 30 °C and 150 rpm. Germinated conidiospores were harvested via centrifugation (4000 rpm, 5 min), washed twice with sterile water, and ground in liquid nitrogen. The ground samples were immediately resolved in a 50% (*v*/*v*) protein extraction buffer (10 mM HEPES pH 7.5, 0.5 mM EDTA, 0.5 mM PMSF, 1 mM DTT, 0.1% (*v*/*v*) protease inhibitor (Sigma, St. Louis, MO, USA)) incubated 10 min on ice and centrifuged 20 min at 12,000× *g* and 4 °C. Total proteins were separated on 15% SDS–PAGE gels and electrotransferred to PVDF membrane according to the manufacturer’s recommendations (Bio-Rad, Hercules, CA, USA). Protein immunodetection was performed for the SDS–PAGE gels according to the standard procedure using a primary antibody recognizing tagRFP (1:3000 dilution; #R10367, Invitrogen, Carlsbad, CA, USA) and a secondary anti-rabbit horseradish peroxidase-conjugated antibody (1:5000 dilution, Sigma, St. Louis, MO, USA). The protein signals were revealed using a chemiluminescence detection kit (Bio-Rad, Hercules, CA, USA). Images were analyzed on a GelDoc Go gel imaging system (BioRad, Hercules, CA, USA).

### 2.8. Crossing Experiments of N. crassa

The fluorescent strains of different mating types were grown on Westergaard’s medium at 25 °C and long-day conditions for 21 days. Mature, ejected ascospores were collected in sterile water and incubated at 65 °C for 30 min to inactivate the contaminating macroconidiospores. Ascospores were incubated on a thin VMM + S medium at 30 °C for 6 to 12 h until germination.

### 2.9. Microscopy

Fungi were incubated on thin agar plates. At different time points, the fungi in different developmental stages were cut out of the plates, transferred to optical slides, covered using sterile water and coverslips, and investigated microscopically.

The yeast cells grown in the fluid medium were harvested via centrifugation at 5000 rpm for 10 min and then washed three times and resuspended in a glucose-containing minimal medium prior to microscopical analysis.

Microscopic analyses were performed with an epi-fluorescence microscope (ECLIPSE Ci, Nikon, Tokyo, Japan) equipped with a GigE camera (Imaging Source, Bremen, Germany). Epi-fluorescence filter blocks TRITC (bandpass, excitation 540/25 nm, dichroic mirror 565 nm, and barrier filter 605/55 nm) and GFP (longpass, excitation 480/40 nm, dichroic mirror 505 nm, and barrier filter 510 nm) were used for red and green fluorescence detection, respectively. The images were analyzed using Nikon software NIS elements D basic.

Confocal fluorescence analyses were performed using a confocal laser scanning microscope (Leica, TCS SP5). The fusion constructs with eGFP were excited at 488 nm, and emission was detected at 500–550 nm, while the tRFP-containing fusion constructs were excited at 543 nm, and emission was detected at 590–610 nm. The images were analyzed with the Leica LAS AF Lite 2.7.3.9723 software.

### 2.10. Quantification of Images and Statistical Analyzes

Quantification analyses were performed with the ImageJ bundled with 64-bit Java8 software. The length of the hyphae was defined as the half perimeter of the hyphae including the macroconidial (ascosporal) part. The hyphae of different developmental stages were divided into 5 µm fragments from tips to the macroconidial (ascosporal) parts, and fluorescent spots were quantitatively analyzed in each fragment by ImageJ.

Statistical analyses were performed with the SigmaPlot 12.0 software.

## 3. Results

### 3.1. LSP1 Is an Eisosomal Marker Protein in N. crassa

The LSP1 protein in *N. crassa* is a homolog of the *A. nidulans* PILA, which is a core component of eisosomes (Appendix A). Both *A. nidulans* PILA and *N. crassa* LSP1 are homologous to the yeast core eisosomal proteins PIL1 and LSP1 (56% and 58% similarity, respectively, Appendix A). In *S. cerevisiae*, the absence of PIL1, but not LSP1, leads to reduced eisosome formation with fewer (less than four per cell) and bigger eisosomal patches [5], which demonstrates that PIL1 is required for the proper formation or maintenance of eisosomes [5]. We transformed a *Δpil1* yeast strain [5] expressing the GFP-coupled yeast LSP1 protein (*lsp1:gfp:his*, *Δpil1::kan^R^*) with the *N. crassa lsp1::rfp* (*Nclsp1::rfp*) reporter gene construct to determine if the NcLSP1 localizes to yeast eisosomes and is able to complement the *pil1* knock-out phenotype, hence indicating a functional similarity of NcLSP1 and PIL1.

We induced NcLSP1:RFP formation by adding galactose to induce gene expression and monitored the subcellular localization of the fluorescent proteins using CLSM. Without induction, the transformed yeast cells showed only the green fluorescence of the yeast LSP1:GFP fusion protein, localized to a few, large eisosomal patches at the membrane (see Figure 1A). After 16 h of induction, the NcLSP1:RFP signal was detectable and found to highly colocalize with the green fluorescent signals in the eisosomal patches (Figure 1B). We also investigated the number of eisosomes before and 16 h after induction and observed a significant increase in the average number of eisosomes per cell after induction from 6.2 to 8.5, *p* < 0.0001, as shown in Figure 1C. As a positive control, we also complemented the yeast *Δpil1* strain with the yeast PIL1 protein by using the same RFP expression vector as that used in former studies. As shown in Figure 1C, the yeast protein caused a significant increase in the eisosome number, similar to the *N. crassa* LSP1 protein. Moreover, the number of eisosomes in the two complemented strains was not significantly different. Progressive induction for more than 16 h led to the massive accumulation of the NcLSP1:RFP protein in the cytoplasm, making the detection of eisosomal localization almost impossible.

### 3.2. Subcellular Distribution of NcLSP1

*N. crassa* is a heterothallic fungus, with both sexual and asexual propagation [34]. We investigated the subcellular distribution of the NcLSP1::tRFP protein in different developmental stages using CLSM. In macroconidiospores and mature hyphae, no significant cytoplasmic pool of NcLSP1::tRFP was detectable (Figure 2). Our investigations of macroconidial sections in different focal planes showed the accumulation of the NcLSP1 exclusively at the cell periphery in several distinct spots at the plasma membrane (Figure 2Ai). In germinating macroconidia, the NcLSP1 is restricted to plasma-membrane-associated spots (Figure 2Aii–ix), while in mature hyphae, the protein also accumulates at the septi (Figure 2B,C).

During the sexual life cycle, the hyphae of different mating types fuse, and nuclear fusion occurs in the ascus prior to meiosis. Finally, eight ascospores are formed in each ascus [35]. Mature ascospores are thick-walled spores, and the young ascospores accumulate pigment during the maturation process and emerge hyphae to form new colonies [35,36,37]. In young ascospores (Appendix A), the NcLSP1 was found to be localized at the periphery as well as in the cytoplasm, which was confirmed with the fluorescence intensity profile analysis (Appendix A). Many bright fluorescent spots were observed at the periphery of the ascospores, which was similar to the distribution of those in macroconidia. In significant contrast to macroconidia (Appendix A), LSP1 was not only located in the plasma membrane but also in the cytoplasm of ascospores. In the equatorial plane of the ascospores, a significant fluorescence signal was detectable (Appendix A). The fluorescence in the cytoplasm was almost as strong as in the peripheral spots, and the even distribution of the fluorescence signal did not indicate accumulation in the ER or vesicles (Appendix A). Changes in NcLSP1 distribution in the different developmental stages of ascospore germination were also examined (Appendix A). At the first germination phase (Appendix A), one or two round buds grew out of an ascospore at the poles. No fluorescence could be detected in the mature ascospore body because of the thick cell wall and the melanin accumulation, but there was significant fluorescence detectable in the cytoplasm and in peripheral spots at the plasma membrane of the buds. In the cytoplasm of early buds, the signal was evenly distributed and strong. As the buds grew longer (Appendix A), the fluorescence in the cytoplasm became much weaker, and more fluorescent spots were observed at the periphery of the germinating tubes.

### 3.3. Eisosome Density and Distribution in Different Developmental Stages

When comparing macroconidia with hyphae, the density, which is expressed by the spot quantity per unit length, of fluorescent spots in macroconidia was much higher than that in hyphae. Furthermore, the density of the spots in hyphae was different, depending on their developmental stage (Figure 3A). The density of the spots in macroconidia was about 0.90 ± 0.10 spots/µm (*p* < 0.05), while the spot densities in the hyphae of different developmental stages (without macroconidiophores) differed from 0.08 ± 0.08 spots/µm (*p* < 0.05) in 5–10 µm long hyphae to 0.26 ± 0.04 spots/µm (*p* < 0.05) in 25–35 µm long hyphae. The density of eisosomes in the early hyphae, which were below 35 µm in length, obviously increased as the hyphae grew longer and stronger, but when the hyphae were longer than 35 µm, the density divergence became less obvious (Figure 3A). The comparison results of fluorescence intensity between macroconidia and hyphae are shown in Appendix A. After analyzing the single-spot fluorescence intensity and the total fluorescence intensity in macroconidia and hyphae, we found that the spots in macroconidia showed a somewhat stronger fluorescence than the spots in hyphae, while their integrated fluorescence intensity was much more different, which indicates that the different fluorescence intensities between macroconidia and hyphae are mainly caused by the different quantity or density of eisosomes (NcLSP1::RFP) in macroconidia and hyphae.

Interestingly, the distribution of eisosomes was heterogeneous along the hyphae germinating from macroconidiospores (Figure 3B). Especially in young hyphae, the sections closer to the macroconidiospores usually have more eisosomes. The longer the hyphae grew, the more fluorescent spots tended to form close to the macroconidial basis. However, once the hyphae were longer than 60 µm, the differences became insignificant. In the mature hyphae that were observed 24 h after germination, the distribution of eisosomes was homogenous and stable within 20 min of investigation (Appendix A).

The density of the fluorescent spots at the periphery of ascospores (0.33 ± 0.074 spots/µm; *p* < 0.05) was lower than that in macroconidiospores (0.90 ± 0.1 spots/µm; *p* < 0.05) (Figure 3C). Investigations on the eisosome density in the young hyphae germinating from ascospores showed no significant differences in different developmental stages, with 0.22 ± 0.06 spots/µm at 10 to 20 µm hyphae length and 0.24 ± 0.05 spots/µm at 20–30 µm hyphae length, whereas in mature hyphae, the eisosome density was reduced to 0.19 ± 0.03 spots/µm. Interestingly, in the mature hyphae germinating from both asco- and macroconidiospores, the eisosome density was lower than that in the germinating hyphae, but in contrast to the hyphae germinated from macroconidia, there was no increase in the eisosome density during the germination of hyphae from ascospores. Additionally, eisosome distribution in the hyphae originating from ascospores was homogenous, without any accumulation in the older hyphal parts closer to the ascospores. This was in strong contrast to the hyphae germinating from macroconidia. Even when the hyphae grew stronger and started to branch (Appendix A), no polar distribution of eisosomes was detectable.

### 3.4. Eisosome Formation at Hyphal Tips Germinated from Macroconidio- or Ascospores

Intriguingly, eisosomes were found to be absent from the tips of the hyphae germinating from macroconidia. Figure 2 sequentially shows the elongation of the hyphae from macroconidiospores. Among them, Figure 2Ai–v show the early stages of hyphal germination. The buds tended to germinate from the areas of macroconidiospores without any visible NcLSP1 accumulation. Fluorescent spots were clearly present at the junctions of the buds and in the macroconidiospores but not at the tips. As the buds grew bigger during this period, there was no fluorescence detected in the tiny germination tubes. Figure 2B,C show the later stages of hyphal elongation. There was still no NcLSP1 accumulation at the tips of the hyphae. However, occasionally, single spots could be observed, as shown in Figure 2Avii, where one fluorescent spot was detected near the tip of a newly germinated hypha. The fluorescence intensity of the spot was as strong as that of spots in macroconidia, but spots such as these were rarely detected in our study. In contrast to macroconidiospores, the NcLSP1 was also present at the tips of hyphae germinated from ascospores. In Figure 4A, germinating asco- and macroconidiospores are presented. The magnified captures of the hyphal tips showed several eisosomal patches in the ascospore-germinated hyphae (white arrows), but the tips of macroconidiospore-germinated hyphae were free from visible NcLSP1:RFP accumulation. Eisosomes were only present at the tips that germinated from the ascospores in early developmental stages. As the hyphae grew longer (Appendix A), fluorescent spots were observed at the periphery as well as at the tips of the hyphae. Moreover, the hyphal tips of the main hyphae and their branches were examined, and eisosomes were found to be located at the tips (Appendix A). Figure 4B demonstrates the length of the eisosome-free tip area in the different developmental stages of macroconidiospore-germinated hyphae. The quantification analyses showed that the length of the eisosome-free hyphal sections depended on the length of the entire hyphae. In the hyphae with a total length of 10–15 µm, the tip length without detectable eisosomes was 4.0 ± 0.9 µm (*p* < 0.05). As the hyphae grew further, the eisosome-free areas increased in length (from 5.8 ± 0.9 µm to 13.3 ± 1.9 µm, *p* < 0.05). After the hyphae reached 60–75 µm in length, the eisosome-free tip region increased to a maximum length of 14.4 ± 1.3 µm (*p* < 0.05).

In summary, we showed that *Neurospora crassa* LSP1 is an eisosomal protein. Further, we demonstrated that *N. crassa* eisosomes have a rather different distribution in the hyphae germinating from sexual and asexual spores and systematically described the eisosomal distribution characteristics in different types of fungal hyphae.

## 4. Discussion

In *S. cerevisiae*, the LSP1 and PIL1 proteins share 74% of their amino acid sequences and consist mostly of an evolutionarily conserved BAR domain, which is able to bind to cell membranes and plays an important role in membrane bending [16]. To form an eisosome, PIL1/LSP1 heterodimers assemble into a scaffold close to the cell membrane [38,39]. Despite their high sequence homology, LSP1 and PIL1 have distinct functions in yeast [18]. The amino acid sequence of the NcLSP1 in *N. crassa* is homologous to the sequences of the LSP1 and PIL1 proteins in *S. cerevisiae.* In our study, the NcLSP1 was expressed in an *S. cerevisiae* Δ*pil1* mutant. The *N. crassa* protein localized to eisosomes and was able to partially complement the *S*. cerevisiae Δ*pil1* phenotype, indicating a functional homology of the NcLSP1 to *S. cerevisiae* PIL1 rather than LSP1, thus providing conclusive proof that NcLSP1 is a functional eisosomal core protein and, therefore, a suitable eisosomal marker.

Subsequently, we expressed the NcLSP1::tRFP protein construct in *N. crassa* to investigate eisosome formation and distribution in the different developmental stages of the fungus. The eisosomes in *N. crassa* form stable punctate patches at the periphery of spores and hyphae during both life cycles, which are similar to that in *A. nidulans* and *S. cerevisiae* [5,6]. In *A. nidulans*, two homologs of the LSP1/PIL1 proteins, PILA and PILB, are described, where PILA is functionally more convergent to PIL1 [6], which could indirectly indicate a functional similarity of the NcLSP1 to PILA. Indeed, both proteins show discrete punctuate localization patterns in all the developmental stages of the fungi [6,7]. However, it is worth mentioning that the other eisosomal core proteins in *A. nidulans*, PILB, and SURG only colocalize with PILA in mature macroconidio- and ascospores [6,7]. Eisosomes are continuously present during the life cycle also in *C. albicans* and *A. gossypii* [23,40]. Notwithstanding, here, we point out some differences, observed only in *N. crassa*. In the early ascospores of *N. crassa*, an additional, cytoplasmic pool of the NcLSP1 was detected, which is not reported in other filamentous fungi. Moreover, the eisosome density and distribution were different in the hyphae germinating from asexual and sexual spores. The eisosomes in ascospores and the hyphae germinating from ascospores were randomly and homogeneously distributed. The eisosome distribution in macroconidiospores and the hyphae germinating from macroconidiospores had a polar, heterologous pattern, which indicates the different regulatory mechanisms of eisosome formation and/or maintenance between the hyphae germinating from asexually and sexually produced spores. Remarkable differences in growth and intracellular organization between germ tubes and mature hyphae at the apex region in *N. crassa* are already reported [41]. However, the differences between the hyphae germinated from ascospores and macroconidia are poorly investigated. In our study, we discovered structural differences between the hyphae germinated from ascospores and macroconidia.

The fungal hyphae are complex systems with different distinct regions, and each region has its own characteristics concerning structure and function [42]. The hyphal tips are significantly important, as they are associated with polarized growth, stress response, fungal colony growth, and the formation of the characteristic filamentous shape of the mycelium [41,42,43]. Secretory vesicles are actively transported to the growing tips and play roles in the growing process [43], which requires a high degree of plasticity in the growing tip. Eisosomes are stable protein complexes with a scaffold protein base rivet at the cytoplasmic surface of the cell membrane [4,38,44], which play a role in a variety of cellular processes, such as cell wall synthesis and morphogenesis, sphingolipid homeostasis [11,24], and stress response [11]. Furthermore, eisosomes are functionally associated with membrane domain formation [45], nutrient transporter regulation [46], cellular signaling [2], and proton flux organization, and they are important for actin organization and invasive hyphal elongation [11]. Our study showed that, in *N. crassa*, eisosomes were distributed at the tips of the hyphae germinated from ascospores, while they were completely absent from the tips of the germinating hyphae of macroconidia, even though these hyphae are identical in morphology [36]. The hyphae germinated from macroconidia and ascospores were distinct on the cellular level, especially at the hyphal tips, which indicates that physiological processes, such as elongation, polarized growth, and stress response, could also be different in these hyphae. As eisosomes reportedly play role in pathogenicity [10,11], if the differences discovered in *N. crassa* also exist in pathogenic fungi, the mechanism of virulence may also be distinct in the hyphae germinated from sexual and asexual spores, which could provide new strategies for the development of antifungal therapies [4].

## Figures and Tables

**Figure 1 jof-09-00147-f001:**
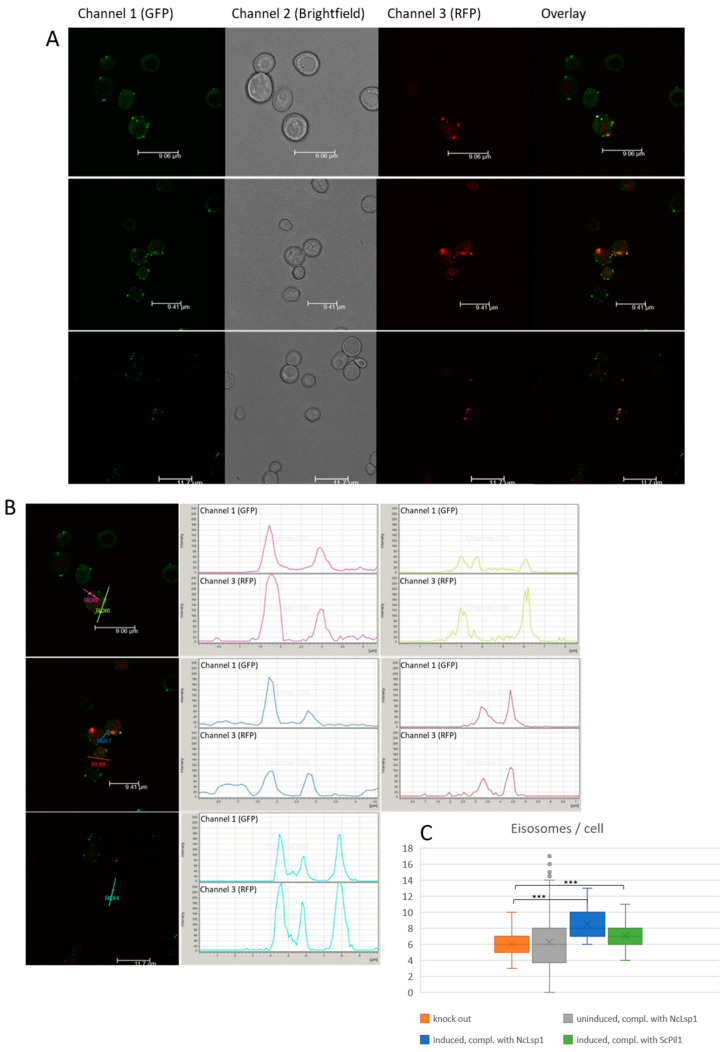
Δpil1 yeast cells complemented by N. crassa lsp1. (**A**) CLSM images of Δpil1 yeast cells expressing the yeast eisosomal marker protein LSP1 coupled to GFP (LSP1:GFP). Yeast cells were transformed with N. crassa lsp1::rfp reporter gene construct. Heterologous expression of NcLSP1::RFP has been induced by galactose. Successfully transformed and induced yeast cells exhibit both green and red fluorescence due to LSP1::GFP and NcLSP1::RFP co-expression. (GFP) Exc.:488 nm, Em.: 500–550 nm; (RFP) Exc.:543 nm, Em.: 590–610 nm (**B**). Fluorescence intensity profiles of single yeast cells from Figure 1A co-expressing S. cerevisiae and N. crassa LSP1 proteins. Intensity of GFP and RFP signals of a selected area (indicated by lines) has been analyzed. The colors of the intensity profiles correspond to the colors of the lines, indicating the analyzed area. (**C**) Number of eisosomes formed in Δpil1 yeast cells (knock out) complemented with N. crassa LSP1 (Nc Lsp1) or S. cerevisiae PIL1 (Sc Pil1). The expression of the complementing proteins was induced by galactose (induced). Expression of N. crassa LSP1 protein significantly recovers the normal eisosome formation of Δpil1 yeast cells, which indicates a functional homology of NcLSP1 to yeast PIL1. Significantly different datasets are indicated in the figure (*** *p* < 0.0001). Complementation with S. cerevisiae or N. crassa protein does not result in significantly different eisosome number.

**Figure 2 jof-09-00147-f002:**
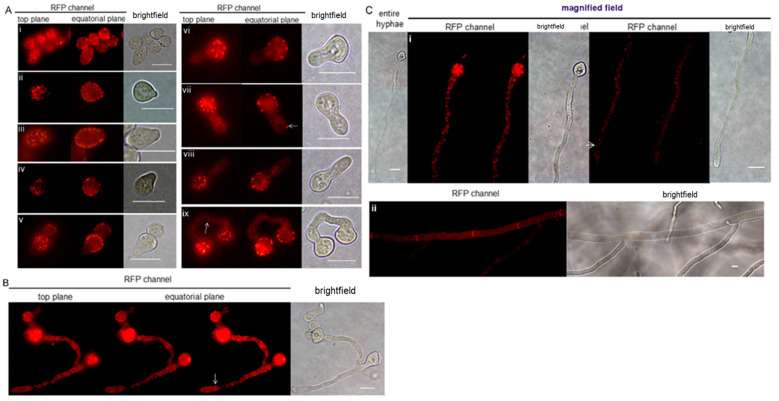
Eisosomes in macroconidia and in hyphae of different developmental stages germinating from macroconidia. CLSM images, Exc.:543 nm, Em.: 590–610 nm. (**Ai**) Images of mature macroconidia. Top and equatorial plane captures are arranged to show the localization of eisosomes. (**Aii**–**ix**) Different developmental stages of the geminating buds growing out of macroconidia. Fluorescent spots located at the buds/germinating tubes are missing; however, rare exceptions are detected (arrows). (**B**) The elongation phase of the germinating hyphae (<100 µm). Fluorescent spots reappear at the plasma membrane, from macroconidial parts to hyphal tips. Fluorescent spots are absent at hyphal tips. The white arrows indicate the fluorescence signal closest to the hyphal tips. (**C**) Fluorescent spot distribution characteristics in long hyphae (>100 µm). The entire germinating hyphae are shown in the bright field. RFP channels show parts of the hyphae. Fluorescent spots are present at most parts of those hyphae, except for the tip parts. (**Cii**) NcLSP1::RFP is enriched at septa of mature hyphae. Scale bars: 10 µm.

**Figure 3 jof-09-00147-f003:**
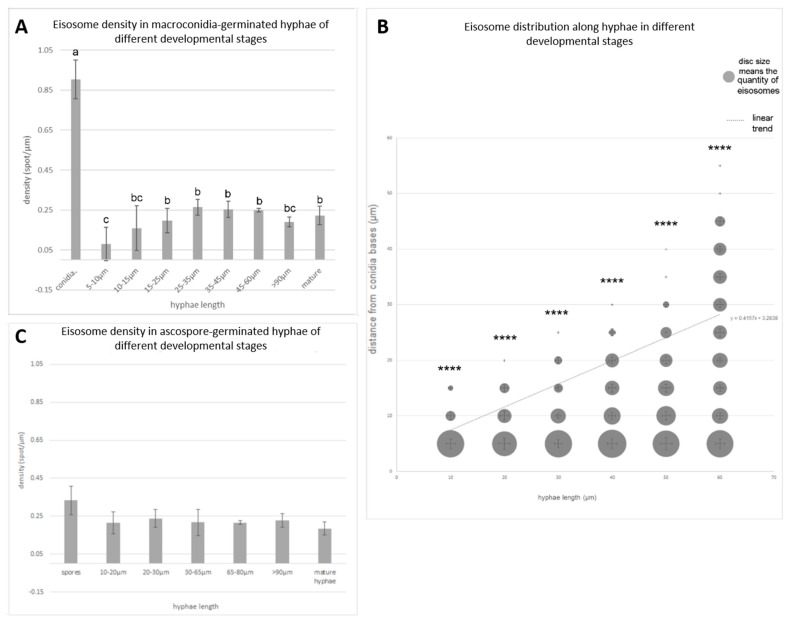
Statistical analysis of eisosome distributions in hyphae germinating from macroconidia and from ascospores. (**A**) The densities of eisosomes distributed in hyphae of different developmental stages (Density is defined as eisosome quantity per µm). Mean ± SD, *n* ≥ 3 experiments, statistical analysis using one-way analysis of variance on ranks (*p* = 1.30 × 10^−10^) with Fisher LSD multiple comparisons test. We mark significant differences between bars with different letters (*p* ≤ 0.05). Same letters indicate no significant difference. (**B**) Eisosome distribution in different areas of hyphae at different developmental stages. The quantity of eisosomes (number/5 µm) was calculated and it is expressed by the area of the discs, which are independent in size from both axes. The corresponding standard deviations are presented as horizontal and vertical bars. The error bars are independent from both x and y axes and correspond to the size of the discs. For each group, mean ± SD, *n* ≥ 3 experiments, statistical analysis using one-way analysis of variance, ****, *p* ≤ 0.0001 (p10 µm = 4.55 × 10^−7^, p20 µm = 2.50 × 10^−7^, p30 µm = 1.16 × 10^−7^, p40 µm = 2.05 × 10^−6^, p50 µm = 1.46 × 10^−6^, p60 µm = 6.31 × 10^−9^). (**C**) The densities of eisosomes distributed in ascospore-germinate-hyphae of different developmental stages. Mean ± SD, *n* ≥ 3 experiments, statistical analysis using one-way analysis of variance on ranks (*p* = 0.076, not significant).

**Figure 4 jof-09-00147-f004:**
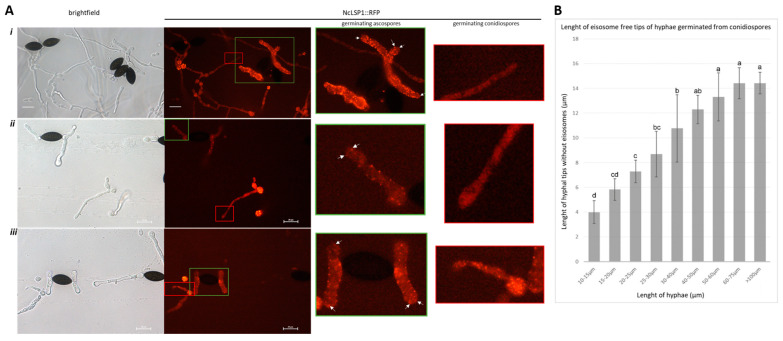
Eisosome formation at hyphal tips germinated from asco- and macroconidiospores. (**A**) CLSM images of germinating asco- and macroconidiospores; Exc.:543 nm, Em.: 590–610 nm. Boxes in the RFP channel indicate the magnified areas on the right side. Colors of boxes and frames of the magnified captures are corresponding (green: ascospore-germinated hyphal tip, red: macroconidiospore-germinated hyphal tip). Arrows indicate eisosome formation at hyphal tips, which exclusively appear in hyphae germinating from ascospores. (**B**) The length of hyphal tips without eisosome formation in macroconidiospores-germinated hyphae of different developmental stages. Mean ± SD, *n* ≥ 3 experiments, statistical analysis using one-way analysis of variance on ranks (*p* = 3.03 × 10^−11^) with Fisher LSD multiple comparisons test. We mark significant differences by the same method as described above (Figure 3A).

## Data Availability

Not applicable.

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
