# Peer review of "Complementation of an Eisosomal Yeast *pil*1 Mutant and Characteristics of Eisosomal Distribution in Hyphae of *Neurospora crassa* Germinating from Two Different Spore Types"

_jof, 2023, doi:10.3390/jof9020147_

Round 1

Reviewer 1 Report

The manuscript describes carefully the distribution and intensity of eisosomes in macroconidia,the germtubes grown from macroconidia and the hyphae developing from macroconidial germtubes in Neurosposra crassa. A comparable developmental series is performed for ascospores, their germtubes and developing hyphae. The eisomes are visualized by tRFP labelled NcLSP1. The latter protein was shown in the manuscript to be homologous to S. cerevisiae  PIL1. The new observation is that in germtubes and hyphae growing from macroconidia (asexual spores) the eisosomes are missing from germtube and hyphal tips while in germtubes and hyphae growing from ascospores (sexual spores) eisosomes ooccr also in the tip region. Since tip region is the most important region for hyphal growth, this is an intriguing observation, although the manuscript, in spite of the careful recordings, do not give any explanation or suggestion if the difference could affect hyphal tip growth. The manuscript is recommended for publication and only few changes are required.

Include the reference Yang,Q The cytoskeleton influences the formation and distribution of eisosomes in Neurospora crassa . Biochem Biophys Res Commun. 2021 Mar 19;545:62-68. doi: 10.1016/j.bbrc.2021.01.037. Epub 2021 Feb 2.

Fig.3. The first sentence in Fig. text should be corrected The statistical analysis include also the distribution of eisosomes from ascospores germinating hyphae.

Condiospore is used as a synonym to macroconidia? This is sometimes confusing. Neurospora has also microconidia? But they are not studied here?

Reviewer 2 Report

Review of Journal of Fungi jof-1918986

Complementation of an Eisosomal Yeast pil1 Mutant and Characteristics of Eisosomal Distribution in Hyphae of Neurospora crassa Germinating from Two Different Spore Types

This study demonstrates that N. crassa lsp1 is functionally homologous to S. cerevisiae pil1. The authors also characterize localization of LSP1 in various cell types during the N. crassa lifecycle and obtain evidence that LSP1 is found in eisosomes.

This is an important contribution to N. crassa biology and filamentous fungi in general.  I have some suggestions to consider during revision.

Major points

1.     Although the authors convincingly demonstrate that N. crassa lsp1 complements the yeast pil1 mutant, they did not rule out the N. crassa pil1 is also functionally homologous to yeast pil1. The latter would require repeating the experiment using N. crassa pil1 for the functional complementation experiments in yeast.

2.     Related to (1), on page 4, top of page 14 and elsewhere, what is the actual % similarity or identity of the yeast proteins to each other, the N. crassa proteins to each other and the pairwise comparisons of N. crassa vs. yeast? Are LSP1 and PIL1 a gene family?

3.     The authors need to perform Western analysis using RFP antibodies to show that the LSP1-RFP fusion protein is expressed in N. crassa and is the correct molecular mass.

4.     Incorporating the analysis in (3) across the different cell types analyzed will also show whether a drop-off in expression of LSP-1 in vegetative hyphae explains the loss of eisosomes observed.  It is known that expression of the ccg-1 promoter is weaker in hyphae vs. spores.

5.     Why is a wild type yeast pil1 control not used for the functional complementation experiments in yeast?  The authors should express the yeast gene under the same promoter and test for functional complementation.

6.     The figure panels in Figure 1 are too small to allow evaluation of the micrographs.

7.     Figure 1C and Lines 205-209: The number of eisosomes under uninduced conditions looks higher than 6.4. The extreme overlap in the error bars suggests that there is not much difference between induced and uninduced?

Minor points

1.     The title to the paper is quite long.  The authors might consider eliminating the phrase regarding functional complementation in yeast and instead focus on the functions in N. crassa.

2.     Line 46: Combine this paragraph with the one above or provide more details about the roles of eisosomes.

3.     Beginning of sentence on Line 157: “Strains carrying fluorescent protein constructs” is a better way to state this.

4.     Line 186: Statistical “analyses”.

Reviewer 3 Report

Formation and distribution of eisosomes as plasma membrane associated protein complexes are not understood so well in filamentous fungi as in model yeast. This manuscript presents experimental data to demonstrate that expression of Neurospora crassa LSP1 (NsLSP1) gene can complement the knockout mutant's phenotype of yeast PIL1, leading to confirmation of NsLSP1 as an eisosomal marker in N. crassa. The authors further demonstrate that N. crassa eisosomes have rather different distribution patterns in the hyphae germinating from sexual and asexual spores, and give systematic descriptions of eisosomal distribution characteristics in different types of fungal hyphae. The presented experimental evidences, interpretations and conclusion provide an in-depth insight into distribution pattern of filamentous fungal eisosomes. I suggest the well written manuscript to be accepted for publication after a minor revision.

Minor suggestion:

All CLSM images shown in Figure 1 remain to be improved for better resolution by cropping and enlarging main cells and enhancing a contrast.

Round 2

Reviewer 2 Report

I appreciated the answer to my points in #1 and 2.  However, the authors did not incorporate the actual % identity or similarity in the text.

Regarding the response to point #3, there are many cases where a fusion protein is cleaved by proteases in vivo.  The fact that the GFP fusion was not cleaved does not mean that this new RFP tagged version is not.

Regarding points 5 and 7, the weak complementation observed for the N. crassa gene in Figure 1C makes checking complementation using the yeast gene under the same promoter imperative.  The two values for eisosomal count under uninduced (presumably gene is not expressed) and induced (assume gene is expressed) may be statistically different, but the actual values are very close--nearly 8 and a little over 8. The presence of the gene does not appear to influence eisosomal count greatly.

Author Response

Comment: Regarding the response to point #3, there are many cases where a fusion protein is cleaved by proteases in vivo.  The fact that the GFP fusion was not cleaved does not mean that this new RFP tagged version is not.

Response: We performed immunodetection of the expressed protein in strain NcT507. The protein size corresponds with the expected size of the fusion protein (ca. 80 kDa). We added the western blot picture as supplementary figure S8. and completed the methods section with corresponding assays.

Comment: Regarding points 5 and 7, the weak complementation observed for the N. crassa gene in Figure 1C makes checking complementation using the yeast gene under the same promoter imperative.  The two values for eisosomal count under uninduced (presumably gene is not expressed) and induced (assume gene is expressed) may be statistically different, but the actual values are very close--nearly 8 and a little over 8. The presence of the gene does not appear to influence eisosomal count greatly.

Response: We conducted the whole experimental procedure with the S. cerevisiae PIL1 as complementing protein. Corresponding changes have been made in the methods section. Data resulted from these experiments have been added to Figure 1C. Results have been modified by the control data the new statistic of the datasets. The complementation results for the N. crassa and S. cerevisiae genes were both highly significant and similar in nature.